# EDCO: Dynamic Curriculum Orchestration for Domain-specific Large Language Model Fine-tuning

## Abstract

Domain-specific large language models (LLMs), typically developed by fine-tuning a pre-trained general-purpose LLM on specialized datasets, represent a significant advancement in applied AI. A common strategy in LLM fine-tuning is curriculum learning, which pre-orders training samples based on metrics like difficulty to improve learning efficiency compared to a random sampling strategy. However, most existing methods for LLM fine-tuning rely on a static curriculum, designed prior to training, which lacks adaptability to the model's evolving needs during fine-tuning. To address this, we propose EDCO, a novel framework based on two key concepts: *inference entropy* and *dynamic curriculum orchestration*. Inspired by recent findings that maintaining high answer entropy benefits long-term reasoning gains, EDCO prioritizes samples with high inference entropy in a continuously adapted curriculum. EDCO integrates three core components: an efficient entropy estimator that uses prefix tokens to approximate full-sequence entropy, an entropy-based curriculum generator that selects data points with the highest inference entropy, and an LLM trainer that optimizes the model on the selected curriculum. Comprehensive experiments in wireless/data communication, medicine and legal domains, EDCO outperforms common curriculum strategies for fine-tuning Qwen3-1.7B/4B and Llama3.2-3B models under supervised and reinforcement learning settings. Furthermore, our efficient entropy estimation reduces computational time by 83.5% while maintaining high accuracy.

## 1 Introduction

Enabling large language models (LLMs) to perform effectively across diverse domains represents a hallmark of machine intelligence (OpenAI, 2023; Google, 2023). Research has recently shifted toward developing domain-specific LLMs, yielding notable applications in fields such as medicine, law, and communication (Wang et al., 2025; Shu et al., 2024; Zhang et al., 2025b). A common approach to constructing such models is fine-tuning a general-purpose pre-trained LLM on specialized datasets. While conventional fine-tuning typically employs random data sampling, emerging evidence indicates that the fine-tuning efficacy is constrained by the training curriculum (Chen et al., 2025), i.e., the order in which training samples are presented. *This is particularly critical for fine-tuning domain LLMs because high-quality domain data is typically scarce and costly*.

However, most existing curriculum learning (CL) strategies for LLMs rely on static data ordering, determined based on heuristic metrics such as difficulty or perplexity (Kim & Lee, 2024). Such fixed curricula remain unchanged throughout training, failing to adapt to the model's evolving ability and knowledge acquisition dynamics, limiting potential gains in both efficiency and final performance. A notable exception is SEC (Chen et al., 2025), which introduces a learnable curriculum policy for curriculum selection. Nevertheless, it suffers from instability because of training the curriculum policy as a bandit problem.

To address these limitations, we propose Entropy-based Dynamic Curriculum Orchestration (EDCO) method, which continuously adapts the training curriculum to the model's evolving learning status. EDCO is grounded in two key ideas: *inference entropy* as a measure of sample impact, and *dynamic curriculum orchestration*. Inspired by recent findings that maintaining high inference

entropy during training provides beneficial learning signals (Cui et al., 2025), EDCO prioritizes samples that maximize inference entropy throughout the training process. This ensures that the model is consistently exposed to data points that challenge its current capabilities and reduce uncertainty most effectively. The EDCO framework integrates three core technical components: (1) *an efficient entropy estimation module*. Due to the computational costs for sweeping over the whole dataset, EDCO uses only prefix tokens to approximate the full-sequence entropy, clearly reducing computational overhead; (2) *a dynamic curriculum generator* that constructs training batches by selecting instances with the highest estimated inference entropy at each training stage; and (3) an *LLM fine-tuning* model for optimizing the LLM. We evaluate EDCO extensively under communication, medical and legal domains. The experimental results demonstrate that EDCO is compatible with supervised fine-tuning and reinforcement learning-based training methods, consistently improving the performance of various types of models in domain-specific fine-tuning.

The contributions of this work are summarized as follows. We leverage the critical insight that *entropy collapse* hinders model learning to propose EDCO, a dynamic curriculum framework. By actively orchestrating training samples to maintain high inference entropy, EDCO prevents premature convergence and sustains effective exploration throughout the fine-tuning process. Besides, we propose prioritizing high inference entropy samples in a *reverse curriculum pattern*, departing from traditional "easy-to-hard" curricula (Kim & Lee, 2024), and introduce a novel efficient entropy estimation technique that reduces computational overhead while preserving accuracy. Moreover, we demonstrate extensive validation and broad applicability through experiments across diverse communication tasks, showing consistent performance gains under supervised and reinforcement learning-based fine-tuning paradigms.

## 2 BACKGROUND

### 2.1 PROBLEM FORMULATION AND LLM FINE-TUNING

Consider we have a domain-specific dataset $\mathcal{D} = \{(x, y)_i\}_{i=1}^{M}$ and a pre-trained LLM $\mathcal{M}_\theta$ parameterized by $\theta$. Here, $x$ is the input prompt (typically a question), and $y$ is the target answer. For simplicity, we use $y \sim \mathcal{M}(\cdot|x)$ to denote sampling an answer $y$ from $\mathcal{M}$ given the question $x$. The primary objective is to optimize the LLM to achieve high answer accuracy on an unseen dataset, represented as $\mathcal{D}'$. LLMs are typically pre-trained on large-scale corpora to acquire general linguistic capabilities. To adapt them to specific domains or tasks, a common approach is to perform continual pre-training followed by fine-tuning on domain-specific datasets. Two primary fine-tuning paradigms are supervised fine-tuning (SFT) and reinforcement learning fine-tuning (RLFT).

In SFT, the model is further trained on a curated dataset of input-output pairs specific to the target domain. The objective is to minimize the cross-entropy loss between the model's predictions and the ground-truth labels:

$$\mathcal{L}_{\text{SFT}} = -\mathbb{E}_{(x,y)\sim\mathcal{D}_{\text{SFT}}} \left[ \sum_{t=1}^{T} \log \mathcal{M}_\theta(y_t|y_{<t}, x) \right], \tag{1}$$

where $x$ is the input prompt, $y$ is the target sequence, $T$ denotes the sequence length, $y_t$ denotes the $i$-th token, and $\mathcal{M}_\theta$ is the LLM policy parameterized by $\theta$.

While SFT is effective for instruction following and style adaptation, it relies heavily on the quality and diversity of the labeled data. In contrast, RLFT leverages RL to optimize the model toward a reward signal, which can be more flexible and scalable, enabling the model to explore diverse solutions. The objective in RLFT is to maximize the expected cumulative reward:

$$J_{\text{RL}}(\theta) = \mathbb{E}_{x\sim\mathcal{D}, y\sim\mathcal{M}_\theta(\cdot|x)} \left[ r(y) \right], \tag{2}$$

where $r(y)$ is a reward function that evaluates the quality of the generated sequence $y$, which can be obtained in the form of self-evaluation (Pang et al., 2024), rule-based (Mu et al., 2024), or verifiable (Su et al., 2025) reward. Common RL algorithms for LLMs include Policy Gradient (Sutton & Barto, 1998), PPO (Schulman et al., 2017), and group-based variants like GRPO (Shao et al., 2024).

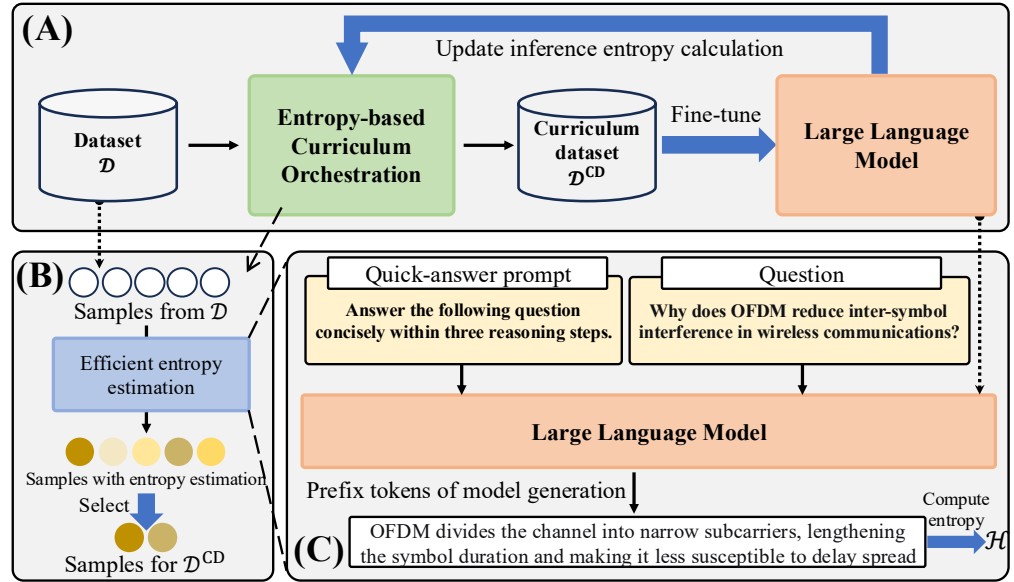

Figure 1: An overview of EDCO method, with three panels: **(A)** Overall training procedure; **(B)** Entropy-based curriculum orchestration module that periodically updates the training curriculum; **(C)** Efficient entropy estimation module that calculates the sample entropy.

## 2.2 THE ROLE OF ENTROPY IN MAINTAINING EXPLORATION

A critical challenge in RLFT is the rapid collapse of the model's inference entropy, often occurring within the first few hundred training steps (Cui et al., 2025). This leads to overconfident models that fail to explore alternative reasoning paths. Empirical studies demonstrate a strong correlation between entropy collapse and performance saturation, modeled by the relationship:

$$R = -a \exp(\mathcal{H}) + b, \tag{3}$$

where $R$ is the validation performance and $\mathcal{H}$ is the LLM's inference entropy. This suggests that performance improvements are effectively "traded" for a reduction in entropy, with the performance upper bound approached as entropy is used up ($\mathcal{H} \to 0$).

Therefore, actively maintaining inference entropy is crucial for sustained exploration and improved generalization. A direct yet promising strategy is prioritizing high-entropy samples during training, ensuring the model encounters challenging examples that hinder entropy collapse and promote ongoing learning.

## 3 METHOD

This section will present our primary contribution, EDCO, a novel training framework that dynamically adapts the curriculum to the model's evolving learning state through periodic inference entropy estimation. Unlike static curricula that follow a predetermined order, EDCO continuously re-evaluates the training sample's significance by calculating the inference entropy on samples with current model. As the overall framework of EDCO shown in Fig. 1, the core procedure operates through: (1) applying an LLM-driven quality filter to exclude low-quality samples; (2) computing inference entropy for the remaining high-quality training samples using our efficient estimation techniques; (3) selecting the top-N highest-entropy samples from this pool to form the training curriculum for the next phase; (4) fine-tuning the model on this selected subset; and (5) repeating the process (2-4) until convergence. This dynamic approach is compatible with both supervised fine-tuning (SFT) and reinforcement learning (RL) paradigms, addressing the critical need for sustained exploration, particularly in RL where entropy collapse can rapidly hinder progress.

## 3.1 LLM-Driven Quality Filter

Before entropy-based sample selection, we apply a crucial quality control step to ensure the integrity of the dynamic curriculum. In LLM fine-tuning, it is paramount to prevent noisy, ambiguous, or incorrect samples from polluting the training process. Our LLM-driven filter evaluates each candidate sample $(x, y)$ across four dimensions: problem clarity, answer accuracy, logical coherence and textual format, and is assigned a score. The quality filtering process produces a refined dataset $\mathcal{D}_{\text{hq}}$. This high-quality subset serves as the foundation for all subsequent entropy calculations and curriculum generation, ensuring that the model learns from challenging yet correct and well-formed examples.

## 3.2 Dynamic Curriculum Orchestration via Inference Entropy

An important motivation for EDCO is that maintaining high inference entropy are more beneficial to training (Cui et al., 2025), as they represent points of maximum uncertainty encourage the model to explore the solution. This constitutes a reverse curriculum strategy that prioritizes more challenging examples rather than following the conventional easy-to-hard progression. EDCO is dynamically adaptive: as the model learns and its uncertainty distribution shifts, the curriculum is updated to reflect new challenging frontiers.

Formally, at training interval $k$, we compute the inference entropy $H(y|x; \theta_k)$ for each sample $(x, y)$ in the training dataset $\mathcal{D}$, where $\theta_k$ denotes the model parameters at interval $k$. The entropy for a given sample is defined as:

$$H(y|x; \theta_k) = -\mathbb{E}_{y \sim \mathcal{M}_{\theta_k}(\cdot|x)} \left[ \log \mathcal{M}_{\theta_k}(y|x) \right]. \tag{4}$$

Samples are then ranked by their entropy values, and the top $N$ with the highest entropy are selected to form the curriculum dataset $\mathcal{D}_k^{\text{CD}}$ for the next interval:

$$\mathcal{D}_k^{\text{CD}} = \{(x, y) \in \mathcal{D} \mid H(y|p_{\text{quick}}, x; \pi_{\theta_k}) \text{ ranks in top } N\}. \tag{5}$$

The model is subsequently fine-tuned on $\mathcal{D}_k$ for a fixed number of steps or until the next curriculum update. This periodic reassessment and selection mechanism ensures the training curriculum remains aligned with the model's continuously evolving capabilities, preventing plateaus and maintaining high learning efficiency throughout the training process.

## 3.3 Efficient Entropy Estimation with Prefix Tokens

A significant challenge in implementing dynamic curriculum is the computational expense of calculating full-sequence inference entropy across the entire dataset. To address this, we introduce two innovative techniques that substantially reduce computational overhead while preserving estimation accuracy.

### 3.3.1 Quick-Answer Prompting

Traditional LLM inference often involves lengthy chain-of-thought reasoning, which is computationally expensive for entropy estimation. We propose *Quick-Answer Prompting* (QAP) technique to modify the input prompt to encourage the model to output the final answer directly without intermediate reasoning steps. Specifically, instead of using a standard instruction like "Solve the following problem step by step", we use a QAP $p_{\text{quick}}$: "*Answer the following question concisely within three reasoning steps*". By *pushing thinking trace towards the answer*, the prefix tokens are more effective in reflecting the model's understanding of the samples, providing a more efficient and concentrated entropy signal. We investigate the effect of QAP in Appendix D.1.

### 3.3.2 Prefix Entropy Approximation

Calculating the exact entropy over the entire output sequence $y$ requires autoregressively generating all tokens and remains prohibitively expensive. We propose *Prefix Entropy Approximation*, which estimates the full-sequence entropy using only the first few tokens of the output. This approach is motivated by the observation that the entropy of the initial tokens strongly correlates with the

---

**Algorithm 1** Entropy-based Dynamic Curriculum Orchestration (EDCO)

---

**Require:** The preprocessed high-quality dataset $\mathcal{D}_{\mathrm{hq}}$, initial model parameters $\theta_0$, prefix length $L$.
 1: $k \leftarrow 0$
 2: **while** not converged **do**
 3:     $\mathcal{D}_k \leftarrow \emptyset$
 4:     **for** each sample $(x, y) \in \mathcal{D}_{\mathrm{hq}}$ **do**
 5:         Construct quick-answer prompt $x' \leftarrow (p_{\mathrm{quick}}, x)$
 6:         Compute prefix entropy $H \leftarrow -\sum_{t=1}^{L} \log \mathcal{M}_{\theta_k}(y_t | y_{<t}, x')$
 7:         Add $(x, y, H)$ to $\mathcal{D}_k$
 8:     **end for**
 9:     Sort $\mathcal{D}_k$ by entropy in descending order
10:     Select top $N$ samples as the curriculum dataset $\mathcal{D}_k^{\mathrm{CD}}$
11:     Fine-tune $\theta_k$ on $\mathcal{D}_k^{\mathrm{CD}}$ for $N$ steps using SFT (Eq. (1)) or RLFT (Eq. (2))
12:     $k \leftarrow k + 1$
13: **end while**
14: **return** $\theta_k$

---

uncertainty of the complete generation. Specifically, we approximate the full-sequence entropy as:

$$H(y|y_{<t}, p_{\mathrm{quick}}, x; \theta_k) \approx -\sum_{t=1}^{L} \log \mathcal{M}_{\theta_k}(y_t | y_{<t}, p_{\mathrm{quick}}, x), \tag{6}$$

where $L$ is a small fixed number of prefix tokens (e.g., $L = 50$). This approximation reduces the computational complexity from $O(T)$ to $O(L)$ per sample, where $T$ is the average output length. In practical, Our experiments demonstrate that this prefix-based entropy maintains a strong rank correlation with the full-sequence entropy, ensuring reliable sample selection while achieving significant speedups.

### 3.4 COMPATIBILITY WITH FINE-TUNING PARADIGMS

EDCO is designed to be agnostic to the underlying fine-tuning algorithm, seamlessly integrating with both SFT and RLFT frameworks. In SFT, the training objective on the selected high-entropy subset $\mathcal{D}_k$ remains the standard cross-entropy loss in Eq. (1). The dynamic curriculum ensures that the model focuses on samples that are currently most challenging, preventing overfitting to easy patterns and promoting broader generalization. In RLFT, EDCO addresses the critical issue of entropy collapse by continuously supplying high-entropy samples that encourage exploration. The RL objective (Eq. (2)) is applied to the selected subset. By maintaining exposed to high inference entropy examples, EDCO effectively delays entropy collapse, allowing the model to explore a wider range of behaviors and discover superior policies.

### 3.5 ALGORITHM SUMMARY

Algorithm 1 summarizes the complete EDCO training procedure. The process begins with the preprocessed high-quality dataset $\mathcal{D}_{\mathrm{hq}}$ (obtained via the LLM-driven filter described in Sec. 3.1) and initial model parameters. At each curriculum update interval, we compute the efficient inference entropy for all samples using quick-answer prompting and prefix entropy approximation. The top $N$ highest-entropy samples are selected to form the training batch for the subsequent phase. The model is then fine-tuned on this subset using either SFT or RL objectives. This cycle repeats until training convergence, ensuring the model is consistently challenged by appropriately difficult examples throughout the learning process.

## 4 RELATED WORK

Our work lies at the intersection of domain-specific adaptation for large language models and curriculum learning. Accordingly, we review relevant literature in three areas.

## 4.1 DOMAIN-SPECIFIC LARGE LANGUAGE MODELS

Recent advances have demonstrated the growing importance of domain-specific LLMs across various professional fields where accuracy, terminology precision, and specialized reasoning are critical requirements (Song et al., 2025; Jeong, 2024; Pal et al., 2024). In medicine, LLMs have evolved from basic information retrieval tools to sophisticated clinical reasoning systems capable of supporting complex diagnostic processes (Berger et al., 2025). Similarly, researchers in the law domain have explored numerous LLMs applications for document analysis, case prediction, and legal reasoning. However, challenges remain in handling complex domain-specific relationships that general models often misunderstand (Colombo et al., 2024). The adaptation of general-domain LLMs for specialized applications in law typically focuses on fine-tuning approaches rather than introducing new architectural innovations (Chen et al., 2024). In communication systems, recent surveys have investigated the integration of LLMs across different network domains, including mobile networks and related technologies, highlighting both opportunities and challenges in this emerging field (Boateng et al., 2024). However, these approaches ignore the training curriculum structure, treating all samples equally valuable regardless of the model's evolving proficiency.

## 4.2 UNCERTAINTY-DRIVEN AND ENTROPY-BASED DATA SELECTION

Uncertainty quantification has become a pivotal metric for evaluating LLM reliability and data quality. Recent works have leveraged entropy and confidence scores for various selection tasks. For instance, Liang et al. (2025) utilize predictive entropy to identify unreliable responses in medical VLMs. Similarly, Zhang et al. (2025a) introduce Long-text Uncertainty Quantification to enhance selective question answering. However, these approaches typically apply uncertainty metrics either as a static pre-filtering step (Liu et al., 2025) or for inference-time control (Agrawal et al., 2025), rather than as a dynamic signal to guide the training trajectory. Unlike active learning methods that focus on selecting unlabeled data for annotation to reduce labeling costs (Xia et al., 2025), EDCO focuses on training efficiency by dynamically re-weighting existing data based on the model's real-time inference entropy, ensuring the model always learns from samples at its capability frontier.

## 4.3 CURRICULUM LEARNING FOR LARGE LANGUAGE MODELS

CL has emerged as a promising approach to improve the efficiency and effectiveness of LLM training. Traditional curriculum strategies often follow an "easy-to-hard" progression, starting with simpler tasks and gradually introducing more complex examples (Kim & Lee, 2024). Some approaches have utilized data distribution characteristics to determine sample ordering, with Static DDCL (Chaudhry & Sharma, 2025) representing an innovative method in utilizing data distribution for curriculum organization. More recently, researchers have begun exploring dynamic curriculum approaches that adapt during training, such as the framework combining CL with LLM reasoning that allows for adaptive adjustment of difficulty levels based on model performance (Zhang et al., 2024b). To overcome the limitations of static curricula, dynamic data selection strategies have gained attention. Hübotter et al. (2024) propose active fine-tuning for test-time adaptation, while Middo (Tang et al., 2025b) introduces a model-informed data optimization loop to enhance fine-tuning quality. Reverse curriculum approaches (Florensa et al., 2017) in reinforcement learning have also demonstrated potential by starting with more complex examples and progressing backward. Yet, these methods remain unexplored for domain-specific LLM fine-tuning. Crucially, no prior work dynamically reorders samples based on the model's instantaneous uncertainty during fine-tuning, an essential factor for maximizing learning efficiency in data-constrained domains.

## 4.4 ENTROPY AS A LEARNING SIGNAL

*Entropy* has gained attention as a valuable signal for guiding LLM training (Tang et al., 2025a). Microscopic Strategy on Responses method (Li et al., 2025) has shown that high entropy in token selection corresponds to greater diversity in training samples, which can make LLM training more robust and less prone to overfitting. Several studies have explored entropy-based data selection techniques to effectively reduce the amount of training data required while maintaining performance (Yin et al., 2024). In RL contexts for LLMs, entropy-based terms have served as robust, self-regularization signals that guide learning without altering the original gradient flow of the base model (Cheng et al., 2025). The EDT method (Zhang et al., 2024a) has also investigated dynamic adjustment of LLM

decoding behavior based on confidence metrics related to entropy. Vocabulary curriculum methods (Yu, 2025) have employed entropy-guided expansion strategies to enable models to learn transferable representations more effectively. Despite these advances, the application of inference entropy as a dynamic curriculum generation mechanism for domain-specific LLM fine-tuning remains largely underexplored, particularly in contexts where SFT and RL training are required.

# 5 EXPERIMENT

In this section, we conduct extensive experiments to verify the effectiveness of EDCO on domain-specific LLM fine-tuning. We conduct experiments across two challenging communication domains, *Data Communication* and *Wireless Communication*, to answer the following key research questions: (1) How does EDCO perform compared to existing curriculum learning methods for LLM fine-tuning (Sec. 5.2)? (2) What is the underlying mechanism of dynamic curriculum orchestration (Sec. 5.3)? (3) How effective and efficient is the proposed entropy estimation module (Sec. 5.4)? (4) How does the prefix token length affect entropy estimation accuracy (Sec. 5.4)?

## 5.1 EXPERIMENTAL SETTING

**Datasets and domains.** We evaluate EDCO mainly on two challenging communication domains: *Data Communication* (Datacom) and *Wireless Communication* (Wireless). We construct a specialized dataset for each domain comprising 20,000 question-answer pairs (filtered to 12,000 high-quality samples) synthesized from a diverse corpus of product documentation, technical solutions, and domain knowledge bases. The datasets encompass diverse question types, including single-choice, multiple-choice, and open-ended QA, covering fundamental principles, product concepts, terminology understanding, and multi-step reasoning tasks. These training dataset could be utilized for SFT and RLFT. All methods are evaluated on a held-out test set of 230 challenging, unseen problems from the same domains. Additionally, we involve datasets from medicine (MedQA (Jin et al., 2021)) and legal (JEC-QA (Zhong et al., 2020)) domains to provide more comprehensive evaluation. We provide more details about the dataset and domain description in Appendix C.1.

**Baseline for comparison.** We compare EDCO against representative baselines with both static and dynamic curriculum learning strategies: (1) **Random Sampling (RS)**: The standard, curriculum-free approach for LLM fine-tuning; (2) **Length-based Curriculum (Length)**: A simple heuristic ordering samples by input sequence length (easy-to-hard based on brevity). (3) **Answer Complexity (AC)**: A heuristic that orders samples by the number of sentences in the answer, representing reasoning depth. (4) **Perplexity-based Curriculum (PPL)**: A classic model-based approach representing the "easy-to-hard" paradigm, where difficulty is determined by a pre-trained model's perplexity (Hu et al., 2024). (5) **Self-evolving curriculum (SEC) (Chen et al., 2025)**: Learn a curriculum policy with UCB algorithm (Auer, 2002) to select the training batch. (6) **Dynamic-PPL**: The dynamic version of PPL method, which updates the curriculum with same interval as EDCO.

**Implementation details.** All experiments are implemented using the MindspeedRL framework (Feng et al., 2025). We use Qwen3-1.7B and Qwen3-4B (Qwen, 2025) as the base LLMs to demonstrate applicability across model scales. For RLFT, we employ the GRPO algorithm (Shao et al., 2024) to train the language models. The rewards are generated by Deepseek-V3 (DeepSeek-AI, 2024) performing automated verification of model responses against ground-truth answers for wireless and datacom domains, and generated from rule-based verification for medical and legal domains. All experiments are conducted on a computing cluster with 256 KUNPENG 920 CPU cores and 8 Ascend 910B3 NPUs. Appendix C.4 lists the hyperparameters used for experiments.

## 5.2 MAIN RESULTS FOR RLFT AND SFT

**Results for supervised fine-tuning.** Fig. 2(A, B) shows SFT results with Qwen3-4B on communication domains. EDCO also achieves the best performance with 33.7% (Wireless) and 36.3% (Datacom) accuracy. Notably, EDCO outperforming PPL methods by 2.0% in Wireless and 3.3% in Datacom, demonstrating the advantage of EDCO over other model-involved CL method. Besides, several "easy-to-hard" baselines (Length, AC, PPL) fail to improve upon or even degrade performance compared to the base model on the Wireless dataset. This reveals a critical pitfall: a poorly designed static curriculum can be actively detrimental to learning, especially in specialized domains

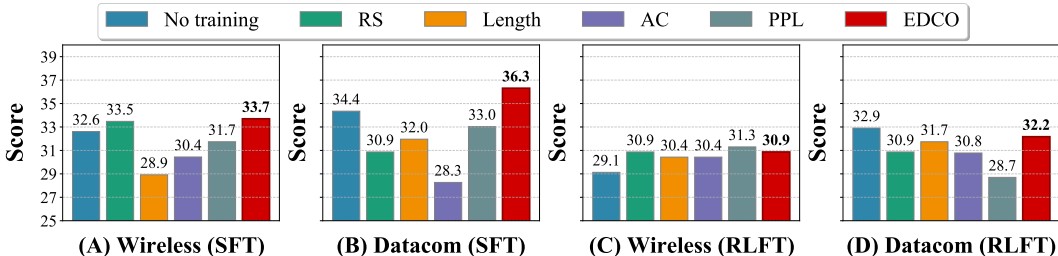

Figure 2: Performance of various fine-tuning strategies on communication domains. The reported results represent the answer accuracy, averaged over three evaluations.

where syntactic simplicity does not equate to conceptual ease. For instance, the poor performance of the AC method in Datacom SFT (28.3%) highlights how a static focus on syntactically complex answers fails to adapt to the model's growing knowledge, leading it to struggle with samples that remain too difficult repeatedly. These results prove that dynamic curriculum orchestration is a more robust and effective strategy across different training paradigms.

**Results for reinforcement learning fine-tuning.** The RLFT setting, shown in Fig. 2(C, D) with the Qwen3-1.7B model, presents a more challenging fine-tuning landscape. In the Datacom domain, most methods struggle to surpass the base model's performance, highlighting the intrinsic difficulty of RL-based alignment in this specialized area. We hypothesis that this is due to there lacks a pre-training step on the domain-specific data to insert relevant knowledge. Even so, EDCO emerges as the top-performing method among all curriculum strategies, demonstrating its robustness even under challenging conditions. On the Wireless dataset, while all methods performed similarly, EDCO remained competitive. The inconsistent performance of the PPL-based curriculum, which underperforms even random sampling in the Datacom domain (28.7% vs. 30.9%), further reinforces the unreliability of static difficulty metrics. In contrast, the relative success of EDCO in this demanding RLFT scenario aligns with our central hypothesis: maintaining high inference entropy provides more robust and effective learning signals, especially when reward signals are sparse or complex.

Table 1: Performance comparison on medical and legal domains using Llama3.2-3B.

| Dataset | No Training | RS | Length | AC | PPL | EDCO |
|---|---|---|---|---|---|---|
| MedQA | 32.1 | 32.9 | 35.1 | 32.4 | 24.6 | **36.7** |
| JEC-QA | 16.2 | 16.2 | 10.5 | 14.6 | 12.4 | **17.4** |

**Results on medical and legal domains.** To demonstrate the generalizability of EDCO beyond telecommunications, we extended our evaluation to two qualitatively different domains using the Llama3.2-3B model (Grattafiori et al., 2024). As the result shown in Tab. 1, EDCO consistently outperforms RS and static curricula (Length, AC, PPL) in these new domains. Notably, on MedQA, EDCO achieves 36.7% accuracy compared to 32.9% for RS and 24.6% for PPL. Similarly, on JEC-QA, EDCO leads with 17.4%. These results validate that prioritizing high inference entropy is a fundamental principle for efficient fine-tuning across diverse fields and is effective across different model architectures (Qwen vs. Llama) and sizes (1.7B to 4B).

**Comparison with dynamic baselines.** To further assess the competitiveness of EDCO at the state of the art, we compared it against two advanced dynamic curriculum strategies on the Datacom domain using the Qwen3-4B model: SEC and Dynamic-PPL. As shown in Tab. 2, EDCO significantly outperforms Dynamic-PPL (47.0% vs. 41.3%). This indicates that the frequency of updates alone is insufficient; the metric used for selection is critical. While perplexity often fails to capture the learning value for fine-tuning, inference entropy effectively identifies the model's capability frontier. Furthermore, EDCO outperforms the bandit-based SEC method (34.78%), which suffered from instability during the policy learning phase in this setting.

Table 2: Comparison against learnable and dynamic baselines on the Datacom domain (Qwen3-4B). EDCO outperforms both the bandit-based SEC and the dynamic perplexity strategy.

| Domain | No Training | RS | SEC | Dynamic-PPL | EDCO |
|---|---|---|---|---|---|
| Datacom | 40.0 | 40.4 | 34.78 | 41.3 | **47.0** |

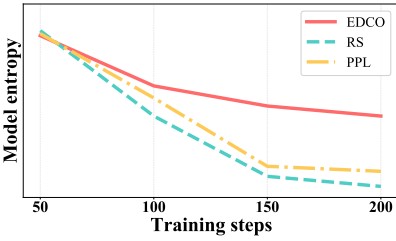

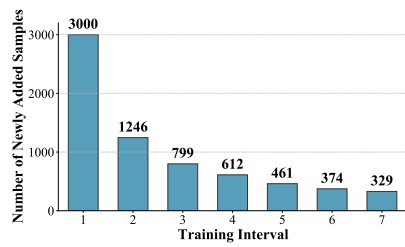

(a) Model entropy during training

(b) New samples at each training interval

Figure 3: Analysis of the training process of EDCO method. **(A)** The model's inference entropy during the training. **(B)** The number of first-time samples added in each training interval.

## 5.3 Analysis of the Training Process

The previous subsection presents that EDCO achieves better performance for domain LLM fine-tuning. Now we investigate the mechanisms behind EDCO's superior performance through detailed training process analysis.

**Entropy change during the training process.** The motivation behind EDCO is to maintain high inference entropy during the training. Fig. 3(a) validates this principle empirically. Specifically, during the training process, we record the model's inference entropy. While the model trained with random sampling sees its entropy decay rapidly, EDCO successfully sustains a high-entropy, high-challenge learning environment throughout training. This demonstrates that our dynamic selection process prevents the model from settling into a low-uncertainty state, constantly pushing it to refine its understanding of more complex or nuanced samples. This sustained challenge directly correlates with its superior final performance.

**Curriculum selection dynamics.** Fig. 3(b) visualizes the composition of the training curriculum at each update interval. The numbers in the figure stand for the number of samples that have never been selected for training previously. The analysis reveals that the curriculum is *constantly evolving*. At each interval, EDCO strategically selects a mix of entirely new, high-entropy samples alongside previously seen samples that remain challenging (i.e., still exhibit high entropy) for the model's current state. This dynamic ensures that complex concepts are not prematurely discarded but are revisited until mastered, while simultaneously introducing new challenges to broaden the model's knowledge. This adaptive "re-challenge" mechanism is a key differentiator from static curricula, which follow a rigid, one-pass sequence.

## 5.4 Effectiveness of Entropy Estimation & Ablation Study

We further verify the effectiveness of the entropy estimation in EDCO through two dimensions: *estimation accuracy* and *computational efficiency*.

**Accuracy of Prefix-based Estimation.** Fig. 4(a) compares the entropy calculated using only a 128-token prefix against the entropy from the full sequence. The results reveal a strong positive correlation, with a *Pearson coefficient of 0.63*. This result is significant: it confirms that the prefix-based approximation serves as a reliable alternative for full-sequence entropy, validating our approach to reduce computational cost without decreasing the integrity of the curriculum signal.

**Computational efficiency.** As detailed in Tab. 3, the efficiency gains of prefix-based estimation are substantial: it reduces the per-sample estimation time from 2.24s to just 0.37s—an 83.5% reduction in computational overhead. This dramatic speedup transforms dynamic curriculum generation from a computationally prohibitive concept into a practical, scalable strategy. Furthermore, when

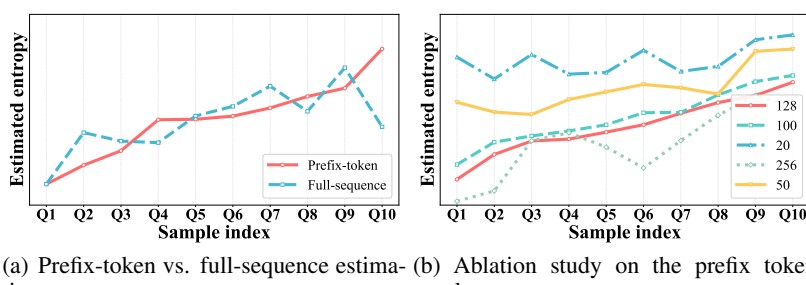

(a) Prefix-token vs. full-sequence estimation

(b) Ablation study on the prefix token number

Figure 4: Analysis of the efficient entropy estimation module in EDCO. **(A)** Comparison of entropy estimation using a 128-token prefix versus the full sequence. **(B)** Ablation study on the prefix length. To better visualize data trends, the sample indices are arranged in ascending order of the entropy estimated with the 128-token prefix. Ablation study on quick-answer prompt is in Appendix D.1.

parallelized across 8 NPUs, the estimation time plummets to 0.04 seconds per sample, making near-real-time curriculum updates feasible even for large datasets. Note that the wall-clock time for EDCO's RL training for 500 steps is 33.58 hours, compared to RS's 30.67 hours (~10% difference). The overhead of entropy estimation is dwarfed by the RL training process itself. Thus, the entropy estimation introduce acceptable overhead for the training.

Table 3: Computational efficiency of entropy estimation methods (seconds per sample).

| Method | Single process | Parallel with 8 cards |
|---|---|---|
| Full-sequence | 2.24 | 0.24 |
| Prefix-based (Ours) | **0.37** | **0.04** |

**Effect of prefix token number.** We conduct an ablation study on prefix token length to provide practical implementation guidelines, with results in Fig. 4(b). The analysis shows that while a very short prefix (e.g., 20 tokens) can lead to unstable estimations, the entropy trends stabilize significantly for prefixes of 50 tokens or more. This indicates that a prefix length of 50-128 tokens strikes an optimal balance between estimation stability and computational efficiency, offering a robust default configuration for future applications.

# 6 CONCLUSION

This work addresses the challenge of efficiently specializing LLMs for specific domains, where data scarcity demands maximally effective fine-tuning strategies. We propose EDCO, a novel framework that introduces a dynamic, entropy-driven curriculum to continuously align the training process with the model's evolving learning state. The key contribution lies in shifting away from static curricula by prioritizing samples with maximum inference entropy, which is efficiently implemented via quick-answer prompting and prefix-based entropy approximation. Extensive experiments in communication domains demonstrate that EDCO consistently enhances performance under both SFT and RLFT paradigms. However, there are still some limitations. First, the efficiency of entropy estimation, while improved from prefix token approximation, still introduces periodic computational overhead compared to standard fine-tuning. Future work could explore more lightweight techniques to predict sample significance without full forward passes. Second, the current method operates on a fixed update interval for curriculum orchestration. An adaptive scheduling mechanism, triggered by performance threshold or entropy convergence, could further optimize the training dynamics. Finally, while we demonstrate effectiveness in communication tasks, the generalizability of EDCO to a wider array of domains (e.g., low-resource languages or highly technical scientific fields) warrants further validation. We believe these interesting directions are worth further exploration for developing more powerful and efficient domain-specific LLMs.

## ETHICS STATEMENT

We adhere to the ICLR Code of Ethics. Our work introduces a method for improving the efficiency of fine-tuning LLMs on domain-specific data. The datasets used in our experiments (from the communications domain) were based on public technical standards and synthetic data, containing no personal information. However, the ethical implications of any model built with our method are contingent on the underlying data and its application. Practitioners should ensure their training data is responsibly sourced and mitigate potential biases. The technique itself is neutral but could be misused; we therefore advocate for its responsible application in alignment with domain-specific ethical guidelines.

## REPRODUCIBILITY STATEMENT

To ensure the reproducibility of our work, we have made substantial efforts to provide all necessary resources and implementation details. We openly the implementation details used in our experiments in Sec. 5. Additionally, the full source code and curriculum generation scripts for EDCO will be made publicly available upon acceptance. Detailed experimental settings, including dataset descriptions, evaluation protocols, hyperparameters, and prompts are provided in the appendix. We believe these materials will facilitate the replication of our results and support future research in dynamic curriculum learning for domain-specific LLM fine-tuning.

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

# Appendix

## Table of Contents

## A  THE USE OF LARGE LANGUAGE MODELS

In the research presented in this paper, LLMs are utilized for two main purposes:

- **Experimental core**: The core contribution of this work involves the fine-tuning of LLMs. Specifically, we developed and evaluated our proposed dynamic curriculum learning framework, EDCO, by fine-tuning the general-purpose Qwen3-1.7B and Qwen3-4B models on specialized datasets from the communication domain.
- **Writing assistance**: An LLM (not Qwen3) was used solely as a tool for polishing the language of this manuscript. It assisted in improving grammar, sentence fluency, and word choice. The model did not contribute to the research ideation, methodology, experimental design, or analysis. The authors take full responsibility for the entire scientific content, findings, and assertions of the paper.

## B  DISCUSSIONS ABOUT REVERSE CURRICULUM LEARNING

This work's prioritization of high-entropy samples represents a departure from the traditional "easy-to-hard" CL paradigm, effectively creating a "reverse curriculum." This section provides additional justification for this choice in the context of fine-tuning pre-trained LLMs within specific domains.

Traditional CL is inspired by human's education process, where a learner starts with foundational concepts and gradually progresses to more complex topics. This is effective when training a model from scratch, as it stabilizes the initial learning stages and prevents divergence caused by overly challenging samples.

However, fine-tuning a pre-trained LLM presents a fundamentally different scenario. The goal of fine-tuning is not to teach the model basic concepts but to specialize its existing knowledge for a specific domain. In this context:

- **"Easy" samples offer diminishing returns.** Samples that the model can already answer with low uncertainty (low entropy) are largely redundant with its pre-existing knowledge. Training on them provides a weak learning signal (i.e., small gradients) and does little to refine its domain-specific abilities.
- **"Hard" samples are most informative.** Samples with high inference entropy are precisely those where the model's general knowledge is insufficient or conflicts with domain-specific nuances. These are the points of highest uncertainty and, therefore, the greatest potential for information gain. By focusing on these samples, EDCO ensures that each training step is maximally efficient at reducing the model's domain-specific predictive uncertainty.

Thus, for domain specialization, a dynamic curriculum that consistently presents the most challenging material, as measured by the model's current state, is more effective than a static, easy-to-hard progression. EDCO's approach ensures the model is perpetually operating at the frontier of its competence, accelerating its adaptation to the target domain.

## C  MORE EXPERIMENT DETAILS

### C.1  MORE DETAILS ABOUT THE EVALUATION DOMAINS.

**Wireless.** The wireless domain is a key branch of information technology, focusing on the research, development, and application of wireless communication technologies. This domain covers a variety of technologies and products, such as 5G, 4G, Wi-Fi, Bluetooth, and NFC, which together build a modern wireless communication infrastructure that supports a variety of application scenarios from mobile communications and the IoT (Internet of Things) to smart homes. 5G technology provides higher data transmission rates, lower latency, and higher connection density, supporting large-scale IoT applications and HD video transmission. Wi-Fi technology is widely used in homes, offices, and public places to provide high-speed wireless network connections.

**Datacom.** The data communication domain is a vital branch of information technology, focused on the efficient and secure transmission of data. This domain includes various technologies and prod-

ucts, such as network devices like routers, switches, firewalls, and associated software solutions. Together, these technologies and products form the foundation of modern network infrastructure, supporting a wide array of applications, from home broadband connections to enterprise-level data centers. The field of data communication is constantly evolving, with emerging technologies like SDN (Software-Defined Networking) and NFV (Network Functions Virtualization) driving transformations in network architecture, enhancing flexibility and manageability.

**Evaluation protocol.** To develop LLM tailored for the communication field and enhance their performance and applicability in the datacom and wireless domains, we construct SFT datasets for both domains. Leveraging a rich set of original corpora, including product documents and solutions from the datacom and wireless domain, we use the LLM synthesis method to generate 20,000 data samples for each domain. These datasets encompass various formats, including single-choice, multiple-choice, and True/False questions, addressing training requirements such as basic principles, product concepts, terminology understanding, and multi-point knowledge reasoning.

In addition, in order to better evaluate the model capability, high-quality evaluation datasets of two domains are constructed. In the first step, the original corpus is collected and obtained through manual quality inspection. The second step is to use the LLM to synthesize objective questions such as selection and judgment based on these corpus. In the third step, the LLM and manual quality inspection are used to review and filter the synthesized objective questions from key dimensions such as question integrity and answer accuracy. Finally, 230 evaluation data samples are constructed for each domain.

### C.2 PROMPTS USED IN EXPERIMENTS

We provide the prompts used in our experiments as follow.

- **Prompts for quick answer**: Please answer the following question in no more than three sentences. If necessary reasoning is required, please minimize the reasoning process as much as possible. Problems: `\{problems\}`

- **Prompts for Single-choice question**: For the following single-choice question, there is only one correct answer. Please analyze the questions and answer options and place the correct option number in `\boxed{}`, for example, `\boxed{A}`.

- **Prompts for Multi-choice question**: For the following multiple-choice question, there are only multiple correct answers. Please analyze the questions and answer options and place the correct option numbers in `\boxed{}`, for example, `\boxed{A,B,D}`.

- **Prompts for True/False question**: For the following question, analyze whether the description in the questions is correct or incorrect, and put the final answer in `\boxed{}`. If the answer is correct, output `\boxed{correct}`. If the answer is incorrect, output `\boxed{error}`.

### C.3 EXAMPLES OF THE DATASET

Tab. C.3 presents the examples from the datasets used for training and evaluation in our experiments.

### C.4 HYPERPARAMETERS

The hyper-parameters for implementing EDCO and experiments are presented in Tab. 5. When implementing baseline methods, we use the same hyper-parameters as EDCO.

## D ADDITIONAL EXPERIMENTAL RESULTS

### D.1 ABLATION STUDY ON QUICK-ANSWER PROMPTING

As shown in Sec. 5.4, our prefix-token entropy estimation correlates well with full-sequence estimation. This accuracy is critically supported by the use of Quick-answer Prompting (QAP). We conducted an ablation study to demonstrate its importance.

| Domain | Example question | Target answer |
|---|---|---|
| Wireless-training | "What should I do if the DSP GTP-PATH command output shows that the GTP path is in the DEETECT state when the alarm is generated?" | "...1.First, check whether the peer GSN address specified in the alarm information is valid...;2.Next, execute the PING command to check if the link is normal...;3.Confirm whether the peer GSN can respond to ECHO messages..." |
| Wireless-testing | "During the deployment and operation of the LMT, the LMT is forcibly connected in HTTPS or WSS mode to ensure secure connection. In this mode, digital certificates are required for authentication. In addition, ...In such a deployment scenario, if the MAE of a carrier is set to HTTP login mode and the LMT is set to forcible HTTPS connection mode, what will happen? A. ...B. If the LMT connection mode is set to Force HTTPS, MAE proxy login fails to access the LMT due to protocol mismatch. The connection cannot be established even if the OM channel is normal. C. ...D. ..." | "The answer is " |
| Datacom-training | "When configuring the Segment VXLAN feature, how can you enable EVPN as the VXLAN control plane on Transit Leaf1 and Transit Leaf2, and configure BGP EVPN peer relationships?" | "...1.Enter the BGP view or BGP multi-instance view...;2.Enter the BGP-EVPN address family view...;3.Configure the split group for BGP EVPN peers (groups)...;4.Enable the function to mark routes received from BGP EVPN peers as re-originated..." |
| Datacom-testing | "...The device supports creating subinterfaces on Layer 2 Ethernet and Layer 2 Eth-Trunk interfaces for VLAN termination to achieve inter-VLAN forwarding. However, the USG9500 series devices do not support creating subinterfaces on these two types of interfaces." | "The answer is error" |

Table 4: Examples from the datasets used in our experiments.

Table 5: Hyper-parameters for training EDCO and baselines.

| Hyper-parameters | Value |
|---|---|
| Prefix token Num. | 128 |
| Epoch Num. | 2 |
| Batch Size | 8 |
| Learning Rate | $1.25e-6$ |
| Learning Rate Decay Style | $cosine$ |
| Train Iterations | 3000 |
| Sequence Length | 4096 |
| Actor Learning Rate | $1e-6$ |
| Actor Learning Rate Decay Style | $constant$ |
| Gamma | 1 |
| Lambda for RL | 0.95 |
| Mini Batch Size | 4 |
| Clip Ratio for RL | 0.2 |

The purpose of QAP is to push the model to begin generating the substantive part of its answer within the prefix window (e.g., the first 128 tokens). Without QAP, the model might use the entire prefix to simply rephrase the question or begin a lengthy preamble, delaying the actual answer. In such cases, the prefix entropy would only reflect the model's uncertainty about the question's phrasing, not its uncertainty about the underlying answer, making it a poor proxy for sample difficulty.

As shown in Fig. 5, when QAP is removed, **the Pearson correlation coefficient between prefix-based and full-sequence entropy drops significantly from 0.63 to 0.32**. This confirms that QAP is essential for making the prefix-token entropy a reliable and effective signal for our dynamic curriculum.

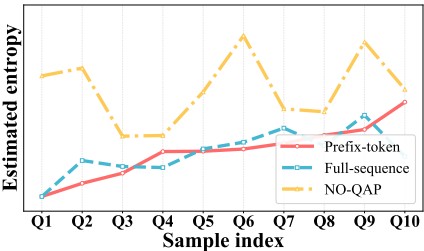

Figure 5: Ablation study on the quick-answer prompting (QAP). Prefix-token entropy estimation has a strong correlation with the full-length estimation (see blue and red lines in the figure).

### D.2 CURRICULUM DIFFICULTY FROM DIFFERENT CL METHODS

To better understand the behavior of different CL strategies, we analyzed the intrinsic difficulty of the samples selected by each method at the beginning of training. We measured difficulty by evaluating the base model's accuracy (Qwen3-1.7B before any fine-tuning) on the first batch of samples chosen by each curriculum.

As shown in Table 6, EDCO and AC select the most difficult samples, with the base model achieving very low accuracy on them. However, AC's extremely low accuracy also shows that answer length fails to indicate the problem difficulty in this setting. In contrast, Length-based and Perplexity-based curricula select comparatively easier samples.

### D.3 CURRICULUM ORCHESTRATION WITH MODERATE-ENTROPY WINDOW

To investigate whether selecting the samples with highest entropy arise from nonsensical edge cases or OOD errors, which is harmful to model optimization, we conduct a "Moderate-entropy" experiment. Instead of selecting the top-N highest entropy samples (Top 0-6.67%), we selected a "Mod-

Table 6: Base model accuracy on the initial training batch selected by different curriculum strategies. Lower accuracy indicates a selection of harder, more informative samples for the pre-trained model.

| Method | Accuracy |
|--------|----------|
| EDCO   | 20%      |
| Length | 38.75%   |
| AC     | 3.75%    |
| PPL    | 31.25%   |

Table 7: Experiments with moderate-entropy for sample selection.

|       | No training | RS    | Top 5-11.67% | Top 0-6.67% |
|-------|-------------|-------|--------------|-------------|
| Score | 40          | 40.43 | 44.78        | 46.96       |

erate" window (e.g., Top 5-11.67%) and compared the fine-tuning performance. We conducted experiments on Datacom domain with Qwen3-4B, as the results shown in the Tab. 7.

The original EDCO outperform the Moderate-entropy strategy by 2.18%. We attribute this robustness to the LLM-driven quality filter (Sec. 3.1) in EDCO's pipeline. Because the filter for logical coherence and correctness before entropy ranking, "nonsensical" high-entropy outliers are removed early. Consequently, the remaining high-entropy samples represent legitimate "hard" examples (frontier knowledge) rather than noise, validating the effectiveness of the reverse curriculum strategy.

