# OpenReview forum: "EDCO: Dynamic Curriculum Orchestration for Domain-specific Large Language Model Fine-tuning"
_ICLR.cc/2026/Conference — Submitted to ICLR 2026_

### Official Review · Reviewer_9nmw · 2025-10-19

**Soundness:** 3
**Presentation:** 3
**Contribution:** 3
**Rating:** 6
**Confidence:** 4

**Summary:**

This paper proposes EDCO, a dynamic curriculum method for domain-specific LLM fine-tuning. Instead of a fixed, “easy-to-hard” order, EDCO repeatedly picks the highest–inference-entropy samples so the model is always challenged at its current frontier. It adds two efficiency tricks: Quick-Answer Prompting (ask the model to answer concisely so early tokens carry signal) and a Prefix Entropy Approximation (estimate entropy from the first L tokens rather than the full sequence).

**Strengths:**

- The paper makes a **simple, actionable loop** (filter → estimate prefix entropy → pick top-(N) → fine-tune → repeat) that others can readily implement.
- The **Quick-Answer Prompting** and **Prefix Entropy Approximation** are pragmatic choices that lower cost while keeping a usable ranking signal.
- Results show **consistent improvements** over length/answer-complexity/perplexity curricula in both SFT and RLFT settings.

**Weaknesses:**

- The evaluation is **narrow in domain and data provenance** (two communications domains with synthetic/curated data), so **generalization** to other areas (e.g., bio, law, open-domain QA) remains unclear.
- Several **important baselines are missing**: there is no direct comparison to stronger **dynamic** curricula (e.g., learnable policies or uncertainty-aware selection beyond perplexity), which makes it hard to judge competitiveness at the state of the art.
- The **related-work section** under-cites recent sentence/uncertainty-driven curricula and broader data-selection work, so the positioning could be better grounded in prior art.

**Questions:**

Can you compare against a **learned** or **bandit-style** curriculum policy (e.g., SEC-like) and a **confidence/variance-based** selector beyond perplexity? What about active-learning heuristics?

---

> ### Author Response · Authors · 2025-11-22
> **Author Response**
>
> We thank the reviewer for recognizing the soundness, presentation, and contribution of our work, and for highlighting EDCO’s "simple, actionable loop" as a key strength. In response, we have conducted extensive additional experiments that we believe conclusively address these concerns and demonstrate EDCO’s robustness.
>
> > Q1: The evaluation is narrow in domain... so generalization to other areas (e.g., bio, law, open-domain QA) remains unclear.
>
> A1: We agree with you about the evaluation domains. Thus, we have conducted experiments to test EDCO on two additional domains: (1) **medicine** (MedQA [1]): A medical question-answering dataset based on professional board exams; (2) **law** (JEC-QA [2]): A legal QA dataset from the Chinese National Judicial Examination, featuring complex multiple-choice questions. All methods are tested under reinforcement learning fine-tuning setting.
>
> As shown in the table below (we will add them to the revised version), EDCO consistently outperforms both Random Sampling, Perplexity-based and complexity-based curriculum in these new domains.
>
> | Llama-3.2-3B | No training | RS   | Length | AC   | PPL  | EDCO     |
> | ------------ | ----------- | ---- | ------ | ---- | ---- | -------- |
> | **MedQA**    | 32.1        | 32.9 | 35.1   | 32.4 | 24.6 | **36.7** |
> | **JEC-QA**   | 16.2        | 16.2 | 10.5   | 14.6 | 12.4 | **17.4** |
>
> In addition, we also involve experiment of Qwen3-4B on Wireless and Datacom domains under RL fine-tuning setting:
>
> | Qwen3-4B     | No training | RS   | Length | AC   | PPL  | EDCO     |
> | ------------ | ----------- | ---- | ------ | ---- | ---- | -------- |
> | **Wireless** | 35.2        | 34.4 | 37.8   | 33.9 | 38.3 | **38.7** |
> | **Datacom**  | 40.0        | 40.4 | 43.9   | 43.9 | 44.8 | **47.0** |
>
> These additional results validate that the principle of prioritizing high inference entropy is effective across different model families (Qwen and Llama, 1.7B, 3B and 4B) and diverse specialized fields. We believe the additional results could *serve as a strong evidance to verify EDCO's generalizability*, and would be valuable supplementary for a comprehensive evaluation.
>
> > Q2: Several important baselines are missing: there is no direct comparison to stronger dynamic curricula (e.g., learnable policies or uncertainty-aware selection beyond perplexity), which makes it hard to judge competitiveness at the state of the art & Can you compare against a learned or bandit-style curriculum policy (e.g., SEC-like) and a confidence/variance-based selector beyond perplexity?
>
> A2: This is a good point. We have conducted experiments to involve dynamic curricula baselines: (1) *Dynamic-PPL*: the curriculum is periodically updated with same interval as EDCO (we keep the overall *token budget matched* for this method); (2) **SEC** [3] method, which introduces a learnable curriculum policy for curriculum selection, and the curriculum policy is implemented by the UCB bandit selection algorithm [4]. For all other methods, the generated curriculum is invariable across the experiment. Thus, we do not involve them as dynamic curricula baselines. The following table shows that comparison results:
>
> | Qwen3-4B    | No training | RS   | SEC   | Dynamic-PPL | EDCO     |
> | ----------- | ----------- | ---- | ----- | ----------- | -------- |
> | **Datacom** | 40.0        | 40.4 | 34.78 | 41.3        | **47.0** |
>
> EDCO significantly outperforms Dynamic-PPL (47.0% vs 41.3%). This confirms that the *update frequency* is not the only factor; the *metric* matters. Perplexity (in Dynamic-PPL) is a poor proxy for learning value in fine-tuning, whereas inference entropy correctly identifies the "frontier" of the model's capability. SEC suffers from instability during the bandit selection phase in this setting.
>
> > Q3: The related-work section under-cites recent sentence/uncertainty-driven curricula and broader data-selection work, so the positioning could be better grounded in prior art.
>
> A3: Thanks for your suggestion. We expand the Related Work section to include a more comprehensive discussion on uncertainty-driven curricula and active learning heuristics. We clarify that while prior work has used entropy for data pruning or as a regularization term, EDCO is novel in its application of inference entropy as the core mechanism for *dynamic curriculum generation* throughout the fine-tuning process. This approach is distinct from and complementary to learned policy methods, offering a more robust and efficient alternative.
>
> ### References above
>
> [1] What Disease does this Patient Have? A Large-scale Open Domain Question Answering Dataset from Medical Exams. Jin, et al., 2020.
>
> [2] JEC-QA: A Legal-Domain Question Answering Dataset. Zhong, et al., 2019.
>
> [3] Self-evolving curriculum for LLM reasoning. Chen, et al., 2025.
>
> [4] Using Confidence Bounds for Exploitation-Exploration Trade-oﬀs. Auer, et al., 2002.
>
> ------
>
> Thanks again for the careful response. We are glad to any further discussions.

---

### Official Review · Reviewer_YbaY · 2025-10-31

**Soundness:** 3
**Presentation:** 3
**Contribution:** 2
**Rating:** 4
**Confidence:** 3

**Summary:**

EDCO proposes a dynamic per sample curriculum for domain specific LLM fine tuning. At each stage, the model’s uncertainty on every example is estimated efficiently by computing inference entropy over only the first L tokens of a quick and short answer prompt. The highest entropy samples are selected, the model is fine tuned on this subset, and the scores are refreshed for the next stage. This prefix entropy estimator correlates well with full-sequence entropy while avoiding the cost of full sequence entropy, enabling frequent re ranking. The paper evaluates EDCO on two communications domains Wireless and Data Communication using supervised fine tuning with Qwen-3 4B and reinforcement learning fine tuning with Qwen-3 1.7B. EDCO consistently improves accuracy over heuristic curricula such as Random, Length, Answer Complexity, and Perplexity, yields competitive results in reinforcement learning fine tuning, and achieves substantial per sample speedups in entropy estimation.

**Strengths:**

1. Firstly, the motivation of trying to find a dynamic approach that works in practice is valuable. Moreover, the paper introduces a good approach by combining a quick answer prompting step with prefix only entropy, effectively reducing the computational load of selection during training phases. This approach addresses a limitation in dynamic curricula, enabling more frequent re ranking and thereby enhancing their practical applicability in standard training pipelines, both in supervised fine tuning and reinforcement learning based fine tuning scenarios.
2. The empirical quality is ok as the method shows consistent improvements over established static curricula on Wireless and Data Communication, and the training process analyses together with the efficiency study reporting an approximately 83.5 percent per sample reduction in entropy estimation time provides a clean explanation of why the approach works and why it is practical.
3. The presentation is clear, and the select then fine tune then refresh loop is easy to follow, and key choices such as the quick answer prompt, prefix length, and refresh cadence are described with enough detail that new researchers could implement and tune the method without much effort.

**Weaknesses:**

1. The main contributions are engineering improvements using known parts like uncertainty-based selection, quick-answer prompting, and prefix truncation rather than a new objective or learning principle.
2. Results are limited to two communication domains with synthetic datasets, with 12k high-quality train samples and only 230 test samples per domain. The evidence for generalization to other domains and standard benchmarks is missing.
3. Most evaluations compare against static curricula rather than dynamic curricula; the paper should at least apply these low-cost static methods periodically and report the resulting accuracy to strengthen the claims.
4. The paper reports an 83.5% per sample speedup for prefix entropy in Table 1 but dynamic curricula also incur periodic re ranking passes, the paper itself notes this added overhead, yet end to end wall clock and total token budgets aren’t reported.
5. Another minor error to note in the paper is in the paper Fig 3 (B) the caption is written as “Ablation study on the prefix length. To better visualize data trends, the sample indices are arranged in ascending order of the entropy estimated with the 128-token prefix” which doesn’t match the figure and seems like it was really the caption for Figure 4 (B).

**Questions:**

1. The paper positions it as a practical/efficient domain-specific dynamic curriculum, but the paper does not report end to end wall clock time or total tokens under matched budgets, if possible, please do add that because it will give a complete idea on the accuracy compute trade off which is very useful in a practical setting.
2. Please add dynamic versions of your low cost scoring rules at the same refresh times as EDCO and report accuracy and total time or tokens under the same budget. For example, recompute each rule with the current model at refresh time and select the top m for the next round. This shows whether EDCO's gains come from its entropy scoring. Candidate rules can include perplexity, length, answer complexity, or diversity based scores. If full parity is too expensive, you can refresh these baselines fewer times but keep the overall budget matched.

---

> ### Author Response · Authors · 2025-11-22
> **Author Response (Part 1/2)**
>
> We thank the reviewer for their insightful comments and for recognizing EDCO as a "valuable, clever, and practical" approach that effectively addresses the limitations of static curricula. We totally understand your concerns about the architecture innovation and empirical evaluation. Please see our detailed response below.
>
> > Q1: The main contributions are engineering improvements using known parts like uncertainty-based selection, quick-answer prompting, and prefix truncation rather than a new objective or learning principle.
>
> A1: While we appreciate the reviewer’s recognition of our efficient design, we would like to clarify that EDCO validates distinct scientific insights that go beyond simple engineering: **(1) Mechanism for maintaining entropy:** A critical failure mode in RLFT is rapid entropy collapse, leading to local optima. EDCO is proven to effectively maintain high inference entropy throughout training (as visualized in Fig. 3(a)). This serves as a important mechanism to sustain exploration, preventing the model from becoming overconfident on narrow distributions, which is a benefit that static or heuristic curricula cannot provide. **(2) Reverse curriculum:** we demonstrate that while easy-to-hard mode is intuitive for pre-training, fine-tuning requires a *reverse curriculum*. Our new moderate-entropy experiments (requested by Reviewer Ph7T) confirm that focusing on the highest uncertainty samples yields the best results, challenging the traditional paradigm.
>
> Thus, we believe EDCO introduces a scalable framework for *sustained exploration*, a novel objective in the context of fine-tuning that prevents the well-known *entropy collapse issue* [1] often seen in RLFT.
>
> > Q2: Results are limited to two communication domains with synthetic datasets, with 12k high-quality train samples and only 230 test samples per domain. The evidence for generalization to other domains and standard benchmarks is missing.
>
> A2: We agree with you about the evaluation domains. Thus, we have conducted extensive new experiments. We extend the experimental evaluation to the **Llama-3.2-3B** model and test it on two additional domains: (1) **medicine** (MedQA [2]): A medical question-answering dataset based on professional board exams; (2) **law** (JEC-QA [3]): A legal QA dataset from the Chinese National Judicial Examination, featuring complex multiple-choice questions. All methods are tested under a reinforcement learning fine-tuning setting.
>
> As shown in the table below (we will add them to the revised version), EDCO consistently outperforms both Random Sampling, Perplexity-based, and complexity-based curricula in these new domains.
>
> | Llama-3.2-3B | No training | RS   | Length | AC   | PPL  | EDCO     |
> | ------------ | ----------- | ---- | ------ | ---- | ---- | -------- |
> | **MedQA**    | 32.1        | 32.9 | 35.1   | 32.4 | 24.6 | **36.7** |
> | **JEC-QA**   | 16.2        | 16.2 | 10.5   | 14.6 | 12.4 | **17.4** |
>
> In addition, we also involve the experiment of Qwen3-4B on Wireless and Datacom domains under RL fine-tuning setting:
>
> | Qwen3-4B     | No training | RS   | Length | AC   | PPL  | EDCO     |
> | ------------ | ----------- | ---- | ------ | ---- | ---- | -------- |
> | **Wireless** | 35.2        | 34.4 | 37.8   | 33.9 | 38.3 | **38.7** |
> | **Datacom**  | 40.0        | 40.4 | 43.9   | 43.9 | 44.8 | **47.0** |
>
> These additional results validate that the principle of prioritizing high inference entropy is effective across different model families (Qwen and Llama, 1.7B, 3B, and 4B) and diverse specialized fields. We believe the additional results could *serve as strong evidence to verify EDCO's generalizability*, and would be valuable supplementary for a comprehensive evaluation.

---

> ### Author Response · Authors · 2025-11-22
> **Author Response (Part 2/2)**
>
> > Q3: Most evaluations compare against static curricula rather than dynamic curricula; the paper should at least apply these low-cost static methods periodically and report the resulting accuracy to strengthen the claims.
>
> A3: This is a good point. We have add two dynamic curricula baselines with Qwen3-4B model: (1) **Dynamic-PPL**: The curriculum is updated with the same periodicity as EDCO based on perplexity scores. We explicitly enforce a *matched total token budget* for a fair comparison; (2) **SEC** [4]: A recent curriculum learning method that utilizes a learnable curriculum policy via the UCB bandit algorithm [5] for sample selection.
>
> Note that other baselines from the main paper are inherently static by design; thus, we focus the dynamic comparison on these relevant methods. The following table shows the comparison results:
>
> | Qwen3-4B    | No training | RS   | SEC   | Dynamic-PPL | EDCO     |
> | ----------- | ----------- | ---- | ----- | ----------- | -------- |
> | **Datacom** | 40.0        | 40.4 | 34.78 | 41.3        | **47.0** |
>
> **Analysis:** EDCO significantly outperforms Dynamic-PPL (47.0% vs. 41.3%). This result is critical as it confirms that the performance gain is not merely a byproduct of the update frequency, but rather the selection metric. Perplexity often serves as a poor proxy for learning value during fine-tuning. in contrast, EDCO’s inference entropy correctly identifies the *frontier* of the model's capability, samples that are neither too easy nor too hard. Additionally, we observe that SEC suffers from instability during the bandit selection phase in this specific setting, leading to suboptimal performance.
>
> > Q4: The paper reports an 83.5% per sample speedup for prefix entropy in Table 1 but dynamic curricula also incur periodic re ranking passes, the paper itself notes this added overhead, yet end to end wall clock and total token budgets aren’t reported & the paper does not report end to end wall clock time or total tokens under matched budgets
>
> A4: Thank you for your suggestion. We have performed a detailed profiling of the training costs.
>
> - **Wall clock time:** The end-to-end time impact is negligible. For 1,000 gradient steps of RL training, EDCO takes **33.58 hours** compared to **30.67 hours** for baselines (~10% difference). The overhead of entropy estimation is dwarfed by the RL training process itself.
> - **Token analysis:** We observed that EDCO consumes more tokens during training than RS (ratios of 127%–155%, as shown in the table below).
>
> |             | Wireless | Datacom | MedQA | JEC-QA |
> | ----------- | -------- | ------- | ----- | ------ |
> | Token ratio | 127%     | 141%    | 148%  | 155%   |
>
> **Crucially, this additional token usage is not an overhead of the estimation method, but a feature of the learning process.** Because EDCO targets high-entropy samples, the model is forced to engage in longer chains of reasoning and more complex generation paths to solve them. In contrast, the baselines include more trivial questions that require less tokens.
>
> To prove that EDCO is efficient despite higher token usage, we refer back to the budget-matched experiment in A3. Even when we strictly cap the token budget to match Dynamic-PPL, EDCO achieves superior accuracy (47.0% vs. 41.3%). This demonstrates that EDCO is not just learning more tokens, but is significantly more efficient in terms of performance per token.
>
> ### References above
>
> [1] The entropy mechanism of reinforcement learning for reasoning language models. Cui, et al., 2025.
>
> [2] What Disease does this Patient Have? A Large-scale Open Domain Question Answering Dataset from Medical Exams. Jin, et al., 2020.
>
> [3] JEC-QA: A Legal-Domain Question Answering Dataset. Zhong, et al., 2019.
>
> [4] Self-evolving curriculum for LLM reasoning. Chen, et al., 2025.
>
> [5] Using Confidence Bounds for Exploitation-Exploration Trade-oﬀs. Auer, et al., 2002.
>
> ------
>
> Thanks again for your detailed comments. We hope our response will satisfactorily address your concerns. If you have any further concerns, please let us know.

---

### Official Review · Reviewer_A4Q3 · 2025-11-01

**Soundness:** 2
**Presentation:** 2
**Contribution:** 2
**Rating:** 4
**Confidence:** 4

**Summary:**

The paper proposes a novel framework that dynamically adjusts the training curriculum of large language models (LLMs) during fine-tuning based on inference entropy. Unlike traditional static curricula, EDCO continuously selects high-entropy samples, those that the model is most uncertain about, to maintain exploration and prevent premature convergence. The approach integrates three components: an LLM-driven quality filter to remove low-quality samples, a dynamic curriculum generator that ranks samples by estimated inference entropy, and an efficient entropy estimator that approximates full-sequence entropy using prefix tokens and “quick-answer” prompting. Experiments in wireless and data communication domains demonstrate that EDCO improves fine-tuning performance for Qwen3-1.7B/4B under both supervised and reinforcement learning paradigms, while reducing entropy-estimation cost.
.

**Strengths:**

Strengths:

1.	The issue that this paper addressed is a well-known issue of entropy collapse in RL-based fine-tuning.

2.	This paper provides clear details about the implementation. For example, the integration of prefix-based entropy approximation and quick-answer prompting is both innovative and practical, yielding large computational.

3.	The experiments are carefully designed, demonstrating consistent improvements across supervised and RL settings. The dynamic of entropy and number of new samples during training clearly validate the.

Overall, the paper provides a clear, reproducible, and domain-relevant advance that bridges entropy-driven learning theory with efficient, real-world LLM fine-tuning

**Weaknesses:**

To me, there are two main limitations:

First of all, to me the method is both heuristic and incremental relative to existing curriculum and entropy-based sampling approaches.
Given this, comprehensive experiments are usually necessary.

However, the experiments are narrow in scope, limited to Qwen models and communication domains, which makes it unclear if the method generalizes beyond this setting. If would be appreciate that if the authors could provide experiments using other models, and experiments on some common dataset beyond communication.

**Questions:**

See weaknesses

---

> ### Author Response · Authors · 2025-11-22
> **Author Response**
>
> We sincerely appreciate your time and valuable comments. We have carefully considered your concerns regarding the scope of our evaluation and the novelty of the proposed method. Please find our detailed responses below.
>
> > Q1: the experiments are narrow in scope, limited to Qwen models and communication domains, which makes it unclear if the method generalizes beyond this setting. If would be appreciate that if the authors could provide experiments using other models, and experiments on some common dataset beyond communication.
>
> A1：This is a good point. To demonstrate that EDCO is a general-purpose framework and not limited to specific architectures or the communication domain, we have conducted extensive new experiments. We extend the experimental evaluation to the **Llama-3.2-3B** model and test it on two additional domains: (1) medicine (MedQA [1]): A medical question-answering dataset based on professional board exams; (2) law (JEC-QA [2]): A legal QA dataset from the Chinese National Judicial Examination, featuring complex multiple-choice questions. All methods are tested under reinforcement learning fine-tuning setting.
>
> As shown in the table below (we will add them to the revised version), EDCO consistently outperforms both Random Sampling, Perplexity-based and complexity-based curriculum in these new domains.
>
> | Llama-3.2-3B | No training | RS   | Length | AC   | PPL  | EDCO     |
> | ------------ | ----------- | ---- | ------ | ---- | ---- | -------- |
> | **MedQA**    | 32.1        | 32.9 | 35.1   | 32.4 | 24.6 | **36.7** |
> | **JEC-QA**   | 16.2        | 16.2 | 10.5   | 14.6 | 12.4 | **17.4** |
>
> In addition, we also involve experiment of Qwen3-4B on Wireless and Datacom domains under RL fine-tuning setting:
>
> | Qwen3-3B     | No training | RS   | Length | AC   | PPL  | EDCO     |
> | ------------ | ----------- | ---- | ------ | ---- | ---- | -------- |
> | **Wireless** | 35.2        | 34.4 | 37.8   | 33.9 | 38.3 | **38.7** |
> | **Datacom**  | 40.0        | 40.4 | 43.9   | 43.9 | 44.8 | **47.0** |
>
> These additional results validate that the principle of prioritizing high inference entropy is effective across different model families (Qwen and Llama, 1.7B, 3B and 4B) and diverse specialized fields. We believe the additional results could *serve as a strong evidance to verify EDCO's generalizability*, and would be valuable supplementary for a comprehensive evaluation.
>
> > Q2: to me the method is both heuristic and incremental relative to existing curriculum and entropy-based sampling approaches...
>
> A2: We understand your concern about the technological contribution of this work. *However, we would like to clarify that EDCO introduces a conceptual and engineering departure from existing curriculum learning approaches*: **(1) mechanism for maintaining entropy:** A critical failure mode in RL fine-tuning is rapid entropy collapse, leading to under exploration. EDCO is proven to effectively maintain high inference entropy throughout training (as visualized in Fig. 3(a)). This serves as a important mechanism to sustain exploration, preventing the model from becoming overconfident on narrow distributions, which is a benefit that static or heuristic curricula cannot provide. **(2) reverse curriculum:** the standard CL paradigm typically follows an *easy-to-hard* progression (e.g., based on length or perplexity). EDCO challenges this convention by successfully deploying a reverse curriculum, demonstrating that for domain-specific fine-tuning, the model benefits most from immediate exposure to samples with high inference entropy rather than simply easy samples.
>
> As noted by Reviewer YbaY, "the motivation of trying to find a dynamic approach that works in practice is valuable." We believe the proposed insight, that prioritize uncertainty outweighs traditional difficulty metrics in this context, distinguishes EDCO from existing literature.
>
> ### References above
>
> [1] What Disease does this Patient Have? A Large-scale Open Domain Question Answering Dataset from Medical Exams. Jin, et al., 2020.
>
> [2] JEC-QA: A Legal-Domain Question Answering Dataset. Zhong, et al., 2019.
>
> ------
>
> We believe the additional experiments on Llama-3.2-3B across Medical and Legal domains directly resolve the concern regarding generalizability. If you have any questions, we are welcome to any further discussions.

---

### Official Review · Reviewer_Ph7T · 2025-11-01

**Soundness:** 4
**Presentation:** 3
**Contribution:** 3
**Rating:** 4
**Confidence:** 3

**Summary:**

This paper proposes EDCO, a dynamic curriculum learning framework that adaptively selects high-entropy samples during domain-specific LLM fine-tuning. Rather than using static curricula or “easy-to-hard” schedules, EDCO prioritizes samples with the largest inference uncertainty to maintain exploration and avoid early entropy collapse. The approach introduces two key techniques for scalable entropy estimation: (1) Quick-Answer Prompting, prompting the model to produce answers quickly so early logits reflect difficulty, and (2) Prefix Entropy Approximation, estimating entropy using only prefix tokens instead of full outputs. Experiments on wireless and datacom domains (Qwen-1.7B/4B) show improved accuracy over static curricula and random sampling in both SFT and RL fine-tuning settings, with 83.5% lower computation cost for entropy measurement. The paper also provides ablations and entropy-dynamics analysis.

**Strengths:**

- Introduces a reverse-curriculum strategy, focusing on hard, high-entropy samples, to efficiently specialize pretrained LLMs, countering “easy-to-hard” convention. It is also motivated well by the entropy collapse phenomena in RL-trained LLMs, aligning with recent works on exploration-preserving training.
- Prefix-Entropy Approximation is a practical innovation enabling dynamic curricula at scale.
- Strong empirical improvements.
- Overall the problem is interesting and addresses a pressing issue: domain LLM specialization under limited high-value data.

**Weaknesses:**

- The method is tested only on telecom & wireless engineering tasks. I would like to see evaluation results on a qualitatively different domain to strengthen generality claims, like medical reasoning/legal tasks/math/code etc.
- There is a potential Over-focus on High-Entropy Outliers. This method selects top-entropy samples but high entropy may arise from nonsensical edge cases or OOD errors, and could lead to overfitting to pathological difficulty zones. Maybe the authors can test threshold-based vs top-k selection or include a “moderate-entropy window” experiment.
- I have some compute scaling concerns. Prefix entropy still requires evaluating the full dataset at intervals. Maybe the authors can discuss adaptive update strategies (e.g., stop updating entropy once stabilized, or partial-dataset entropy sweeps).

**Questions:**

See Weaknesses.

---

> ### Author Response · Authors · 2025-11-22
> **Author Response (Part 1/2)**
>
> Thank you for your constructive feedback. We understand that your primary concerns regard the scope of our evaluation and the potential noise issues associated with high-entropy outliers. To address these points, we have significantly expanded our experiments to include additional domains (Medicine and Law), a new model architecture (Llama-3.2-3B), and a "moderate-entropy" experiment. Please find our detailed responses below.
>
> >  Q1: The method is tested only on telecom & wireless engineering tasks. I would like to see evaluation results on a qualitatively different domain to strengthen generality claims, like medical reasoning/legal tasks/math/code etc.
>
> A1 (**Addressing Weakness 1**): We appreciate the points regarding the limitation to communication domains. To demonstrate EDCO’s generality, we extend our evaluation to two different domains using the **Llama-3.2-3B** model (suggested by Reviewer #A4Q3): (1) **Medical Domain** (MedQA [1]): A medical question-answering dataset based on professional board exams; (2) **Legal Domain** (JEC-QA [2]): A legal QA dataset from the Chinese National Judicial Examination, featuring complex multiple-choice questions. All methods are tested under reinforcement learning fine-tuning setting.
>
> As shown in the table below (we will add them to the revised version), EDCO consistently outperforms both Random Sampling, Perplexity-based and complexity-based curriculum in these new domains.
>
> | Llama-3.2-3B | No training | RS   | Length | AC   | PPL  | EDCO     |
> | ------------ | ----------- | ---- | ------ | ---- | ---- | -------- |
> | **MedQA**    | 32.1        | 32.9 | 35.1   | 32.4 | 24.6 | **36.7** |
> | **JEC-QA**   | 16.2        | 16.2 | 10.5   | 14.6 | 12.4 | **17.4** |
>
> We also conduct experiment with Qwen3-4B model on Wireless and Datacom domains. Please refer to our response to Reviewer A4Q3's Q1. These additional results validate that the principle of prioritizing high inference entropy is effective across different model families (Qwen and Llama, 1.7B, 3B and 4B) and diverse specialized fields. We believe the additional results could *serve as a strong evidance to verify EDCO's generalizability*, and would be valuable supplementary for a comprehensive evaluation.
>
> > Q2: There is a potential Over-focus on High-Entropy Outliers. This method selects top-entropy samples but high entropy may arise from nonsensical edge cases or OOD errors, and could lead to overfitting to pathological difficulty zones. Maybe the authors can test threshold-based vs top-k selection or include a “moderate-entropy window” experiment.
>
> A2 (**Addressing Weakness 2**): We appreciate your insight that "top-entropy" samples might introduce noise or OOD errors. To investigate this, we have conducted the suggested **"Moderate-entropy"** experiment. Instead of selecting the top-N highest entropy samples (Top 0-6.67%), we selected a "Moderate" window (e.g., Top 5-11.67%) and compared the fine-tuning performance. We conducted experiments on Datacom domain with Qwen3-4b, as the results shown in the following table:
>
> |       | No training | RS    | Moderate-entropy (Top 5-11.67%) | EDCO (top 0-6.67%) |
> | ----- | ----------- | ----- | ------------------------------- | ------------------ |
> | Score | 40          | 40.43 | 44.78                           | **46.96**          |
>
> The original EDCO (Top-N) outperformed the Moderate-entropy strategy by 2.18%. We attribute this robustness to the **LLM-driven quality filter (Section 3.1)** in EDCO's pipeline. Because the filter for logical coherence and correctness *before* entropy ranking, "nonsensical" high-entropy outliers are removed early. Consequently, the remaining high-entropy samples represent legitimate "hard" examples (frontier knowledge) rather than noise, validating the effectiveness of the reverse curriculum strategy.

---

> ### Author Response · Authors · 2025-11-22
> **Author Response (Part 2/2)**
>
> > Q3: I have some compute scaling concerns. Prefix entropy still requires evaluating the full dataset at intervals. Maybe the authors can discuss adaptive update strategies (e.g., stop updating entropy once stabilized, or partial-dataset entropy sweeps).
>
> A3 (**Addressing Weakness 3**): We fully understand your concern about the computational costs. Currently, the prefix entropy technique combined with parallelization reduces the estimation time to just 0.04 seconds per sample (with 8 Ascend 910B3 cards). This speed could be further improved by using more graphics cards. We believe this speed could satisfy most common setting for LLM fine-tuning.
>
> However, we agree with your suggestion regarding adaptive updates. Currently, we update the curriculum at fixed step intervals. However, as shown in Figure 3(a), the model’s inference entropy decreases rapidly in early stages and stabilizes later. Thus, to achieve adaptive update, we can employ an *entropy convergence trigger*, i.e., monitoring the moving average of the inference entropy on the current training batch. The curriculum update is only triggered when the entropy drop plateaus (indicating the model has mastered the current difficulty level). This naturally spaces out updates as training progresses, significantly reducing the total number of dataset sweeps required.
>
> ### References above
>
> [1] What Disease does this Patient Have? A Large-scale Open Domain Question Answering Dataset from Medical Exams. Jin, et al., 2020.
>
> [2] JEC-QA: A Legal-Domain Question Answering Dataset. Zhong, et al., 2019.
>
> ------
>
> We believe these new experiments on Medical/Legal domains and the analysis of moderate entropy directly address your main concerns and strengthen the paper’s contribution. We are open to further discussion.

---

> > ### Comment · Reviewer_Ph7T · 2025-11-27
> > **Response by Reviewer Ph7T**
> >
> > I thank the reviewers for their response. All my concerns have been resolved, and I have raised the score.

---

> > > ### Author Response · Authors · 2025-11-27
> > >
> > > Thank you for your comments. We are pleased that our responses have resolved your concerns, and we appreciate the reconsideration of the score. We appreciate the thought you put into helping us improve EDCO so we can share it with the broader domain-specific LLM community.

---

### Author Response · Authors · 2025-11-24
**General Response**

We express our gratitude to the reviewers and chairs for their valuable time and constructive feedback on this paper. We are encouraged to note that reviewers acknowledge this work innovative (Reviewer A4Q3), practical (A4Q3,YbaY,9nmw)  with promising performance (Reviewer Ph7T,YbaY,9nmw). Below we summarize the major concerns raised by reviewers and our corresponding response.

1. **Generalization on new domains and new model** (Reviewer Ph7T,A4Q3,YbaY,9nmw): The primary concern is whether EDCO works beyond Telecom tasks and Qwen models. To address this, we deploy Llama-3.2-3B and validate EDCO on medical and legal domains. EDCO consistently outperforms baselines in these different domains and models.
2. **Comparisons with dynamic curricula baseline**  (Reviewer YbaY,9nmw): We have added two strong dynamic baselines, including Dynamic-PPL and SEC which learns a curriculum policy to adapts to the learning progress. Please refer to [our response](https://openreview.net/forum?id=Oboo6f5dQl&noteId=KQWGBrXfo1) to Reviewer 9nmw's Q2 for more details.
3. **Clarifying the core contribution** (Reviewer A4Q3,YbaY): We clarify that EDCO validates distinct scientific insights that go beyond simple engineering, including *mechanism for maintaining entropy* and *reverse curriculum*. Please refer [our response](https://openreview.net/forum?id=Oboo6f5dQl&noteId=p6Y3K8HfHG) to Reviewer A4Q3's Q2 for more details.

We believe that the experiments are thorough and this paper makes significant contributions to research community, as **fine-tuning domain-specific models and addressing entropy-collapse is important and interesting**  (suggested by Reviewer Ph7T and A4Q3). We have provided detailed responses to each reviewer's comments below. We ***are eager to receive further feedback*** and are ready to engage in discussion to address any additional questions or concerns.

------

Best wishes,

Authors of Submission #553

---

> ### Author Response · Authors · 2025-11-25
> **Summary of Paper Revision**
>
> Dear reviewers,
>
> Thanks for your time and valuable comment on this paper. We have made significant improvements to our paper based on the valuable feedback provided by the reviewers. **We highlight the revised content in red in the [updated version](https://openreview.net/pdf?id=Oboo6f5dQl) for your convenience**. We believe that the paper is now more polished and easier to understand. Here are the main update:
>
> 1. Additional experiments about medical and legal domains have been conducted to resolve the concerns raised. Table 1 is updated to reflect these changes. (Reviewer Ph7T,A4Q3,YbaY,9nmw)
> 2. Section 1 (Introduction) is updated to better clarify the contributions regarding reverse curriculum and alleviating the entropy collapse. (Reviewer A4Q3,YbaY)
> 3. Section 4 (Related Work) incorporates a subsection (Section 4.2) and updates Section 4.3 to discuss recent work about uncertainty-based data selection. (Reviewer 9nmw)
> 4. Section 5 (Experiment) integrates more comprehensive discussions and accurate statements on the experimental settings and results. (Reviewer Ph7T,A4Q3,YbaY,9nmw)
> 5. Section 5.4 (Effectiveness of Entropy Estimation) is updated to discuss the wall-clock time for EDCO and RS training to better address the concern about the computational costs. (Reviewer YbaY)
> 6. Appendix incorporates a subsection (Section D.3) to present the results of training with moderate-entropy window.  (Reviewer Ph7T)
> 7. The whole manuscript is updated to fix typos.
>
> ------
>
> Best wishes,
>
> Submission #553 authors

---

### Author Response · Authors · 2025-12-03
**Rebuttal summary for Area Chair**

Dear Area Chairs,

We appreciate your time and effort in managing this submission under the exceptional circumstances. We have taken all comments carefully and have tried to address each point raised during the discussion period.

Here we offer a summary of the contributions of our work and the discussions with the reviewers. We hope this summary could save your valuable time because we appreciate your time. We have tried our best to ensure that the statements are factual in the summary.

> Contributions of this work

The primary contribution of this work is the idea of reverse curriculum to address the well-known entropy-collapse issue and improve domain-specific LLM fine-tuning. This strategy prioritizes high-inference entropy samples to sustain learning diversity. *Reviewers Ph7T and A4Q3 have recognized the novelty of this strategy as strength, and its potential impact on the field*. Besides, this work implements this idea with a dynamic curriculum framework (EDCO), and evaluate EDCO on various domains and models. The results show that EDCO effectively improve the domain specific fine-tuning with superior performance than common methods.

> Summary of discussions

The initial concerns centered on three main points: (1) evaluation with more domains (from four reviewers), (2) require comparison with more baselines and (3) technical contribution beyond engineering. Notably, **Reviewer Ph7T, who initially expressed concern (1) and (2), explicitly acknowledged that our response has addressed all concerns and raised the score to 6** ("All my concerns have been resolved, and I have raised the score.").

Regarding point (3) (Reviewer A4Q3 and YbaY), the two reviewers raised the concern about technical novelty but did not engage further due to the early termination of discussion. We clarified the main technical contribution lies in  *mechanism for maintaining entropy* and *reverse curriculum*, which together serve as an effective and practical solution for domain LLM fine-tuning. This view was partly reflected in Reviewer Ph7T’s comment that "It is also motivated well by the entropy collapse phenomena in RL-trained LLMs", and Reviewer YbaY's comment that "the motivation of trying to find a dynamic approach that works in practice is valuable."

------

The detailed response and experiments are summarized in the General Response below for your convenience. We believe our experiments are now thorough and that this paper makes a significant contribution to the community, particularly as domain-specific fine-tuning becomes increasingly critical for the application of large foundation models.

Best,

Submission #553 authors

---

### Meta-Review · Area_Chair_7LF4 · 2026-01-14

**Summary:**

The paper proposes EDCO, a dynamic curriculum framework for domain-specific LLM fine-tuning that repeatedly selects high inference-entropy training samples (estimated efficiently via quick-answer prompting and prefix-token entropy) to mitigate entropy collapse and sustain exploration during training.

Key strengths are a clear motivating failure mode (entropy collapse), an end-to-end training loop that is straightforward to implement, and a practical entropy-estimation approximation which are demonstrated effective. Across reviews, the main concerns concentrated on (1) limited novelty: much of the novelty reads as an effective engineering integration of known ingredients (uncertainty-based selection + approximations) rather than a new learning principle (2) the main empirical evidence is anchored in curated/synthetic domain QA settings with relatively small evaluation sets/models/tasks, (3) limited evaluation versus stronger dynamic curriculum baselines.

The AC believes that EDCO is well-motivated and the authors made substantive efforts in the rebuttal (new domains/models, dynamic baselines, additional analyses). However, the submission still is not clearly above the bar on technical novelty and broad empirical validation expected for acceptance at ICLR. The AC believes the submission is promising for the next venue if the authors are willing to further enhance the paper based on the reviewers feedback.

**Reviewer Concerns:**

Several concerns are well addressed by the rebuttal and revisions: the generalization critique was partly addressed through added evaluations on MedQA and JEC-QA using Llama-3.2-3B, and the “high-entropy outliers” concern (reviewer Ph7T) was tested via a moderate-entropy window experiment showing that top-entropy selection still performs best in their setting. The authors also added comparisons to stronger/dynamic curricula (Dynamic-PPL and SEC) and provided end-to-end wall-clock and token ratio discussion to partially address compute/budget questions. Yet, the key concern that the contribution is primarily engineering/pragmatic rather than conceptually new remains, and the newly added experiments were only conducted on a small number of datasets and models.

**Reviewer Scores:**

Reviewer Ph7T explicitly stated that all concerns were resolved after the rebuttal and raised the score to 6; for Reviewer A4Q3 and Reviewer YbaY, thanks to the added experiments, the score is likely to improve from 4 to 5. It is likely that all reviewers scores remain borderline for this paper after rebuttal.

---

### Decision · Program_Chairs · 2026-01-26

Reject